# Predictive Model of the Risk of In-Hospital Mortality in Colorectal Cancer Surgery, Based on the Minimum Basic Data Set

**DOI:** 10.3390/ijerph17124216

**Published:** 2020-06-12

**Authors:** Juan Manuel García-Torrecillas, María Carmen Olvera-Porcel, Manuel Ferrer-Márquez, Carmen Rosa-Garrido, Miguel Rodríguez-Barranco, María Carmen Lea-Pereira, Francisco Rubio-Gil, María-José Sánchez

**Affiliations:** 1Department of Emergency Medicine, Hospital Universitario Torrecárdenas, 04009 Almería, Spain; 2Instituto de Investigación Biosanitaria ibs.Granada, 18012 Granada, Spain; miguel.rodriguez.barranco.easp@juntadeandalucia.es (M.R.-B.); mariajose.sanchez.easp@juntadeandalucia.es (M.-J.S.); 3CIBER de Epidemiología y Salud Pública (CIEBERESP), 28029 Madrid, Spain; 4Fundación FIBAO, Hospital Universitario Torrecárdenas, 04009 Almería, Spain; molvera@fibao.es; 5Department of General and Digestive Surgery, Hospital Universitario Torrecárdenas, 04009 Almería, Spain; manuferrer78@hotmail.com (M.F.-M.); cirujafrancis@hotmail.com (F.R.-G.); 6Fundación FIBAO, Hospital Universitario de Jaén, 23007 Jaén, Spain; crosa@fibao.es; 7Registro de Cáncer de Granada, Escuela Andaluza de Salud Pública, 18011 Granada, Spain; 8Empresa Pública Hospital de Poniente, El Ejido, 04700 Almería, Spain; maru31es@yahoo.es; 9Department of Preventive Medicine and Public Health, Universidad de Granada, 18071 Granada, Spain

**Keywords:** predictive model, colorectal cancer, epidemiology, public health, mortality

## Abstract

Background: Various models have been proposed to predict mortality rates for hospital patients undergoing colorectal cancer surgery. However, none have been developed in Spain using clinical administrative databases and none are based exclusively on the variables available upon admission. Our study aim is to detect factors associated with in-hospital mortality in patients undergoing surgery for colorectal cancer and, on this basis, to generate a predictive mortality score. Methods: A population cohort for analysis was obtained as all hospital admissions for colorectal cancer during the period 2008–2014, according to the Spanish Minimum Basic Data Set. The main measure was actual and expected mortality after the application of the considered mathematical model. A logistic regression model and a mortality score were created, and internal validation was performed. Results: 115,841 hospitalization episodes were studied. Of these, 80% were included in the training set. The variables associated with in-hospital mortality were age (OR: 1.06, 95%CI: 1.05–1.06), urgent admission (OR: 4.68, 95% CI: 4.36–5.02), pulmonary disease (OR: 1.43, 95%CI: 1.28–1.60), stroke (OR: 1.87, 95%CI: 1.53–2.29) and renal insufficiency (OR: 7.26, 95%CI: 6.65–7.94). The level of discrimination (area under the curve) was 0.83. Conclusions: This mortality model is the first to be based on administrative clinical databases and hospitalization episodes. The model achieves a moderate–high level of discrimination.

## 1. Introduction

Although the incidence of colorectal cancer (CRC) is very high, in recent years its prognosis has improved substantially and mortality has decreased, thanks to advances in surgical techniques and cancer therapies, as well as earlier diagnoses and the use of high quality treatment approaches [1,2].

The rate of in-hospital mortality in patients who undergo CRC surgery is approximately 1.5% [3], according to prospective rather than recorded cohorts. The surgical outcome is multifactorial, influenced both by factors concerning the surgical team and by patient-related variables.

Predictive models have been constructed to estimate the risk of in-hospital mortality following major surgical procedures of the colon and rectum—tumorous or otherwise. Outstanding in this respect are POSSUM [4] (“Physiological and Operative Severity Score for the Enumeration of Mortality and Morbidity”) and derivatives, such as the Portsmouth review, P-POSSUM (Portsmouth-POSSUM) [5], and the model specific for colorectal pathologies, Cr-POSSUM (Colorectal-POSSUM) [6]. The “American College of Surgeons-National Surgical Quality Improvement Program” (ACS-NSQIP) [7], of which several versions have been proposed, is the most commonly used model in the USA. The first model consisted of three sub-models (30-day mortality, severe, and global morbidity). These were subsequently merged into a universal model that provides high quality predictions for CRC. This model has been adapted to many other surgical pathologies with good results [8].

The French Association of Surgery (AFC) has published a four-variable model for estimating mortality risk after major or diverticular colorectal resection. In Spain, the recently conducted Colorectal Cancer Health Services Research (CCR-CARESS) study has contributed significantly to our understanding of these questions [3], recalibrating the above models and assessing their validity for use with a Spanish population. It has been argued that these scores overestimate the risk, especially in patients with elective surgery and in younger patients. This question has been raised with particular regard to the POSSUM and Cr-POSSUM scores [9,10].

Furthermore, the scores proposed to date have been developed on the basis of cohorts of real patients. This provides obvious benefits, but also means that the sample sizes used are limited. No score has yet been developed following the analysis of large numbers of hospitalization episodes, obtained from large clinical administrative databases. Moreover, the reliability of the above scores has not been studied. In addition, the scores currently available use data obtained during hospitalization; however, much of this information is derived exclusively from the surgical area and is not available when the patient is admitted.

In view of these considerations, the aim of the present study is to detect the factors associated with in-hospital mortality in patients undergoing surgery for CRC, based on the Spanish Minimum Basic Data Set at Hospital Discharge (MBDS), thus generating a score based on the variables present when admission takes place.

## 2. Materials and Methods

This observational study is based on a historical cohort of all hospitalization episodes in Spain, of patients aged 20 years or more, classified by codes 153–154 of the International Classification of Diseases, version 9, Clinical-Modification (ICD9-CM), for the period 2008–2014. The unit of analysis is the hospitalization episode (n = 115,841). Only patients with a diagnosis of CRC in the first diagnostic position and who underwent surgery were included. The sample was divided into a training set (TRS) composed of 80% of the episodes, obtained by random selection, and a test set (TES) made up of the remaining 20% in order to perform an internal validation of the model obtained.

### 2.1. Data Sources

MBDS, facilitated by the Health Information Institute of the Spanish Ministry of Health, Consumer Affairs and Social Welfare.

### 2.2. Variables

The main study aim, to predict the risk of in-hospital mortality, was addressed by taking exitus (i.e., death during hospital stay), as the main dependent variable for all the prediction models developed.

The remaining study variables were classed as independent or predictive. The following variables were analyzed: sociodemographic (age, sex, residence), clinical comorbidities (stroke, hypertension, ischaemic heart disease, obstructive pulmonary disease, renal insufficiency, bacterial or aspiration pneumonia), and surgical comorbidities (digestive bleeding, stenosis, anastomotic dehiscence, postoperative ileus, surgical wound infections).

The following variables were also included in the analysis: preoperative hospital stay, readmissions, total number of diagnoses and procedures at discharge (NDD and NPD, respectively), length of stay, and type of admission (urgent vs. scheduled). The NDD was taken as a proxy of the degree and complexity of the comorbidities presented, and the NPD was assumed to be representative of the diagnostic-therapeutic effort. The database was purged for length of hospital stay using a moderate outlier detection procedure according to the classical method in which outliers are defined with the formula T2 = Q3 + 1.5 (Q3 − Q1), where Q identifies the third and first quartiles and T2 is the maximum value of the stay that results from applying the formula.

### 2.3. Statistical Analysis

The analysis was performed on the TRS except for the internal validation process, for which the operation of the final model was evaluated on the TES. All analyses were performed using the SPSS v.17 and Stata/SE v.12 statistical packages.

A descriptive, cross-sectional study was conducted of the main variables: age, sex, stay, type of admission, readmission, comorbidities, NDD, and NPD. The continuous variables are expressed as mean ± standard deviation and the qualitative variables as percentages and frequency distributions.

The association between in-hospital mortality and each of the independent variables was identified by calculating the unadjusted-crude odds ratios (ORu). In addition, three multivariate logistic regression models were constructed from the adjusted odds ratios (ORa), using only the clinical independent variables that were identified on admission and which were statistically significant. The first bivariate logistic regression model included all of the clinical variables that were considered clinically relevant, prioritizing those observed at the time of admission (“Initial Model”, Model 1) [11]. The “intermediate model” was obtained, from which the variables known to cause under-reporting bias, as shown by their low coding rate, were eliminated (Model 2). The third and “final model” constructed, termed the CSMS (Colorectal Spanish Mortality Score), was designed to be as parsimonious as possible (Model 3). The variables were extracted in sequence, according to their degree of contribution to the discrimination of the model.

All three models were evaluated according to their discriminative capacity by the area under the curve (AUC) and according to their calibration, using the Pearson’s χ^2^ goodness-of-fit test, a standardized method based on ungrouped data.

The practical value of the Hosmer–Lemeshow (HL) test, used to evaluate the calibration, is limited by the large sample size [12,13]. Multiple-simulation studies suggest than Pearson’s χ^2^ goodness-of-fit test could be more powerful than the HL test [14]. The discriminative capacities of the intermediate and final models were compared, using the criterion of the maximum extraction of variables with minimum impact on the AUC in the final model.

After applying the final model to the TRS, internal validation was performed by applying the same model to the remaining 20% of the sample (TES) and examining its performance using the holdout method [15]. The AUC values for the two sets were compared using the algorithm proposed by DeLong et al. [16].

Prior to establishing this final model, several alternatives were considered. The variations consisted fundamentally in separating elective and urgent surgery into two different models and in categorizing the age variable. These models provided a poorer goodness-of-fit and, therefore, were discarded.

A predictive mortality score was calculated from the final model, following the method described by Sullivan et al. [17]. The relationship between each individual’s score and the individualized risk was then plotted graphically.

## 3. Results

Of the 92,768 hospitalization episodes evaluated in the TRS, 60.1% (n = 55,735) were male patients and 77.4% were scheduled admissions. All of the analyses and results obtained refer to the period 2008–2014. The patients’ mean age was 69.56 ± 11.67 years and the average stay was 12.50 ± 6.74 days, with an average preoperative stay of 2.51 ± 3.96 days. In each case, 3.77 ± 2.94 medical and/or surgical procedures were performed and 5.96 ± 3.37 diagnoses were made at discharge. In total, 16.3% of the episodes were re-admissions and the in-hospital mortality rate was 4.2%, as shown in Table 1. By location, 24.1% of the tumors were in the rectum, 23.5% in the sigmoid colon, 10.8% in the rectosigmoid junction, and the remainder in the colon.

Laparotomy was performed in 97.22% of emergency surgeries vs. 88.76% of scheduled surgeries, with statistically significant differences (*p* < 0.001). The rate of laparotomies, thus, was higher in emergency surgery.

Bivariate analysis revealed significant differences according to age, which was positively associated with in-hospital mortality (ORu 1.08, 95% CI 1.08–1.09). A higher ORu of mortality in the bivariate analysis was very clearly associated with male sex (ORu 1.17, 95% CI 1.09–1.25), open surgery (ORu 4.17, 95% CI 3.35–5.21), and urgent admission (ORu 5.32, 95% CI 4.99–5.68) vs. scheduled admission, among others. Associated clinical comorbidities included stroke (ORu 2.81, 95% CI 2.33–3.38), anaemia (ORu 1.89, 95% CI 1.73–2.05) and, especially, renal insufficiency (ORu 10.29, 95% CI 9.50–11.16), as shown in Table 2.

Logistic regressions, developed sequentially from the initial model to the final model, as shown in Table 3 and Figure 1, presented a similar discriminative capacity. A comparison of the ROC curves of the intermediate and final models revealed a decrease of 0.007 points in the AUC (*p* < 0.001). In the final model, the variables found to be associated with mortality included age, increased 10-year risk, (OR 1.79, CI 1.78–1.79), urgent admission (OR 4.68, CI 4.36–5.02), COPD (OR 1.43, CI 1.28–1.60), stroke (OR 1.87, CI 1.53–2.29), and renal insufficiency (OR 7.26, CI 6.65–7.94). The AUC was 0.83 (CI 0.82–0.83).

The calibration of the three models was evaluated using Pearson’s χ^2^ goodness-of-fit test. All of the models were statistically significant (*p* < 0.001).

For the internal validation, the final model was applied to the TES cohort (AUC = 0.82). There were no statistically significant differences between this result and that obtained with the TRS cohort (*p* = 0.891), as shown in Figure 2.

The possible scores ranged from 1 to 21 points. The episode with the highest score in the sample produced a score of 17 points. The score–risk correlation, as shown in Table 4, is illustrated in Figure 3.

## 4. Discussion

### 4.1. Findings

Our third, or “final model”, is the first predictive model to be developed in Spain exclusively from clinical administrative databases as a means of stimating the risk of postoperative mortality in patients admitted with CRC. To our knowledge, this is the first model of its type using this source of information. The model is constructed using only the variables present at the time of admission. The results obtained show that this risk estimation instrument provides high discriminative capacity from parameters that are easily and quickly determined.

### 4.2. Comparison with Previous Studies

Many models have been proposed for the context of general abdominal surgery, and others (although fewer) specifically for CRC surgery. One such is the POSSUM score [4], which has been widely used to predict morbidity and mortality in a variety of surgical processes, in addition to being a useful tool for comparison purposes, by adjusted risk [18]. Variations of this model have appeared, such as Portsmouth-POSSUM (P-POSSUM), aimed at correcting the overestimation of mortality risk, which is very common in low-risk patients. P-POSSUM uses the same twelve physiological and six surgical factors to predict postoperative mortality, but applies a different methodology [19]. Both models are applicable to any surgical patient, unlike the one we propose, which was developed specifically for application in patients with CRC.

To adapt these scales for a separate application to different medical specialities, further modifications have since appeared, such as P-POSSUM, Cr-POSSUM, O-POSSUM, and E-PASS [20,21,22,23]. In this context, the Cr-POSSUM model is of particular interest [24].

The Association of Coloproctology of Great Britain and Ireland (ACPGBI) created a system to assess the risk to patients scheduled for CRC surgery. The prognostic mortality rate thus created was validated by Tekkis in a prospective, multicentre study of patients who underwent surgery for CRC [22]. Age, ASA anaesthetic risk, tumor stage, and type of intervention (urgent/scheduled and complete/incomplete) were shown to be independent prognostic factors. Unlike the CSMS, the calculation of the Tekkis risk scale requires the inclusion of pathological and tumor staging variables.

The CR-POSSUM model also considers physiological and surgical variables, but reduces the number of variables required [24], while maintaining the duality of scores (physiological and operative).

In the United States of America, the American College of Surgeons (ACS-NSQIP) model provides accurate estimates and is very useful both for the patient and for the surgeon. It is a relatively extensive model, with some analytical and tumor-related variables and others that are subjective. Overall, the model performs well. It includes variables, such as anaesthetic risk and other analytical variables (albumin, creatinine), together with some relating to the extent of the tumor and indications for surgery. This means that it cannot predict mortality until more complete information is available. In addition, the estimated mortality is established at 30 days after discharge and not during hospitalization. Despite these limitations, the model is acceptable, providing a good level of performance, and could usefully be applied to complement the method we propose.

More recently, European models, such as that of the French Surgery Association (AFC), have appeared. Using the AFC model [25], a study was conducted of 1049 patients undergoing CRC surgery. The following were included as independent factors, assumed to be directly associated with mortality: urgent surgery, loss of > 10% body weight in the last six months, neurological history, and age over 70 years. The authors proposed a simple prognostic mortality model in CRC patients; this model produced a mortality score in the form of points, but the risk was not individualized. This model bears the greatest similarity to the CSMS, since with only four variables it obtains an acceptable estimate of mortality. Nevertheless, it also presents significant drawbacks, including the small size of the sample, the fact that surgery was elective in over 83% of cases and the use of a 70-year cut-off point for the age criterion (thus, the increased risk per additional year of age was not determined).

A study conducted in the Netherlands proposed a mortality prediction model termed the Identification of Risks in Colorectal Surgery (IRCS) [26], which was externally validated in a cohort of Spanish patients. After validation, the model was compared with CR-POSSUM, and was found to achieve a higher predictive power, according to the ROC curve analysis (0.83 vs. 0.76). The POSSUM model, and others derived from it, has been recalibrated in several studies to obtain new logistic coefficients providing more accurate estimates of risk, as has previously been done concerning other non-surgical pathologies [10,27].

In 2015, Kong et al. [28] published a predictive model of in-hospital mortality in patients undergoing colorectal surgery, termed the Colorectal Preoperative Surgical Score (CrOSS). This model was created and validated externally in Australia, and although it also needs to be validated in other contexts, the initial analysis revealed a ROC value of 0.87. It has the advantage of considering only four variables, namely, age, urgent intervention, albumin, and heart failure. In the same year, Walker et al. [29] presented another model (C-statistic 0.80) but the estimates referred to the 90 days after discharge and so the risk estimation was not immediate.

In Spain, the CCR-CARESS study [10] validated and recalibrated the logistic coefficients of several pre-existing models (CR-POSSUM, POSSUM, AFC and ICRS) by reference to a multicentre cohort in 22 Spanish hospitals. This recalibration slightly improved the discriminative capacity of the CR-POSSUM model (from 0.73 to 0.75) and that of the POSSUM model, which rose to 0.77.

Another study [3] used the same cohort to develop a mortality score, and concluded that advanced age (over 80 years), palliative surgery, and chronic obstructive pulmonary disease (COPD) were the factors most strongly associated with mortality. The CCR-CARESS score was first applied to 60% of the sample, and was then validated for the remaining 40%. The three variables cited above were used to create a range of 0 to 5 points, which, in turn, was associated with a given risk of mortality at 30 days. This score is straightforward to apply and has good discriminative capacity, although, unlike the CSMS, it does not evaluate hospital mortality or obtain individualized estimates of risk, referring instead to severity groups.

Thus, almost all the predictive models presented to date take into account the surgery performed and the operative or postoperative variables. In consequence, the risk estimates can only be obtained after the surgery has been performed. Other models provide estimates at 30–90 days or even one to two years after surgery, but not in the immediate postoperative period [30]. The SCMS model (the third or final model) is an auxiliary instrument designed to estimate the risk of mortality in the hospital and before colorectal surgery, together with clinical criteria and other traditional scores. It is intended to assist in decision making and scheduling for patients who need this type of surgery.

### 4.3. Strengths of the Study

In comparison with previously published models, the major contribution of the paper we present is the description of what may be the most radically simplified model yet proposed. It is especially significant that this model is based on the analysis of a very large registry population. Patient age, type of admission, history of COPD, renal insufficiency, or stroke are the independent variables used to predict the risk of mortality before surgery is undertaken. The age variable is present in all of the models previously reported. However, as our model does not take into account the type of surgery performed, but rather the type of admission, the consideration of this variable distinguishes our model from the alternatives. The history of COPD resembles the variable described in other models, such as that of Sluis et al. [26], which includes respiratory failure. Our index includes patient history of stroke as the fourth variable to be considered, and this comorbidity has previously been suggested as a factor that may be associated with mortality [31,32].

In view of these considerations, we believe that the proposed model has adequate discriminative capacity and provides good visual calibration of the deciles of risk. To our knowledge, with one exception, the Spanish population has only been used in the external validation of models for other nationalities, not in the creation of scores. The exception is Quintana et al. [30], but their estimates only considered mortality at one and two years. Our model enables the pre-surgery risk to be estimated and does not require the consideration of complex variables. This model facilitates the provision of more personalised medicine and surgery by estimating the risk faced by each patient individually upon admission.

Finally, although the risk score is derived from the logistic model, certain explanations should be provided. Firstly, the conversion of this score, as shown in Table 4, into risk ranges by quartiles provides estimates with sharp variations. For this reason, the results were softened and the abrupt changes produced by categorization were attenuated, by means of a curve showing the points generated in the score, with the individualized risk for each value, as shown in Figure 3. This alternative can be obtained rapidly and avoids the need for complex calculations. On the other hand, the consideration of clinical variables and those related to patient fragility seems to be more strongly related to mortality in the case of in-hospital mortality. Variables referring to a later stage, such as those related to surgery or the condition of the tumor, have less weight in the model, thus facilitating the very early provision of risk estimates.

### 4.4. Potential Limitations

One of the main difficulties of the proposed model is the question of the relevance of the HL test [13], which is commonly used to evaluate the model calibration. This test is based on a Chi-square test and is therefore affected by elevated sample sizes [33]. In this regard, we preferred Pearson’s χ^2^ goodness-of-fit test, applied to ungrouped data, in view of its possible greater power. Nevertheless, statistically significant results were observed [14].

Another limitation that should be taken into account is the use of hospitalization episodes, rather than individual patients. As a result of this, overfitting might be present. This factor might also account for the low contribution of the gender variable, which was ultimately removed from the model.

A number of important surgical variables (ileus, deshicences, etc.) were not included due to a lack of significance in the elaboration of the third or final model, and this might seem a striking limitation. On the one hand, this absence of variables in the multivariate model is, in fact, a strength, since it allows us to estimate patient risk at the time of admission. However, the exclusion of these factors must also be viewed as a potential limitation, in the sense that they might be relevant to mortality during the first surgical hospitalization. Nevertheless, on balance, we believe that, if the patients had been followed up for three or more months after admission, any such impact from this source would have been observed.

Finally, we must acknowledge the existence of under-registration, a bias that could provoke the appearance of paradoxical and sometimes falsely protective results. This type of bias has been extensively studied by Jencks et al. [34] and research has confirmed that it frequently affects the Spanish MBDS [35]. In the current study, conditions, such as hyperlipidaemia, hypertension, and obesity, present this type of effect. Another aspect related to the characteristics of this type of medical record is the fact that it is very difficult to differentiate an intrahospital complication from a previous comorbidity. However, our final model does, in fact, isolate the variables contributed by the patient at the time of admission.

### 4.5. Implications

This paper describes a new model for estimating the risk of in-hospital mortality. The results obtained raise important considerations regarding disease prognosis and management. A better understanding of individualized risk will allow treatment programs to be adapted accordingly and diagnostic-therapeutic tests streamlined to determine this risk. The model has important implications for improving the quality of health care and may have a significant impact on what has been termed “personalised medicine”. The variables addressed can be obtained in the first few minutes of patient care. Forthcoming studies of recalibration and external validation will ensure the absence of overfitting and will underpin the reliability of this approach to the reality of hospital mortality from CRC.

Finally, it should be stressed that the predictive model proposed is an auxiliary tool that can (and we believe should) be used in conjunction with other clinical-surgical parameters and even with other previous scales. The model is not intended to replace, but rather to complement the consideration of clinical-surgical criteria in the risk assessment process.

## 5. Conclusions

Our study shows that it is possible to create a logistic model and a scoring system to estimate the risk of death of patients undergoing surgery for colorectal cancer. The model obtained is built on variables that place more emphasis on the frailty of the patient than on the intraoperative variables; it also has the advantage of using variables obtained at an early stage in the clinical assessment.

Finally, we reiterate the importance of the role played by clinical variables and comorbidities in predicting the mortality that occurs during the hospitalization of these patients.

## Figures and Tables

**Figure 1 ijerph-17-04216-f001:**
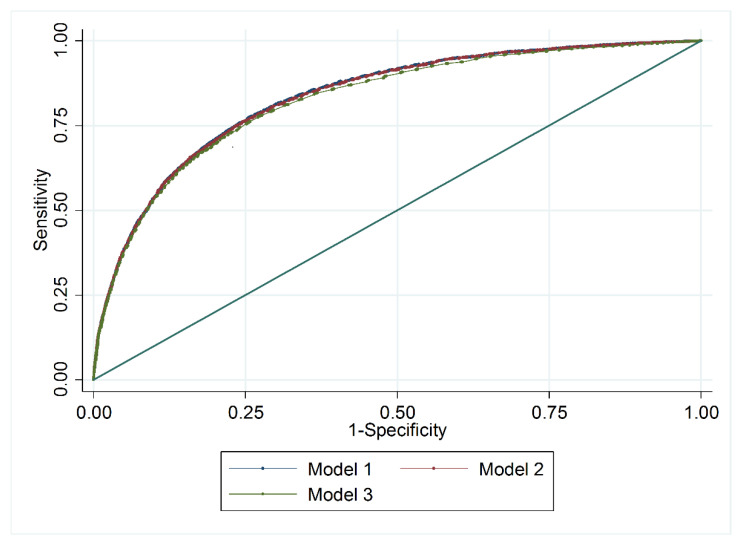
A comparison of AUC slopes for intermediate and final models.

**Figure 2 ijerph-17-04216-f002:**
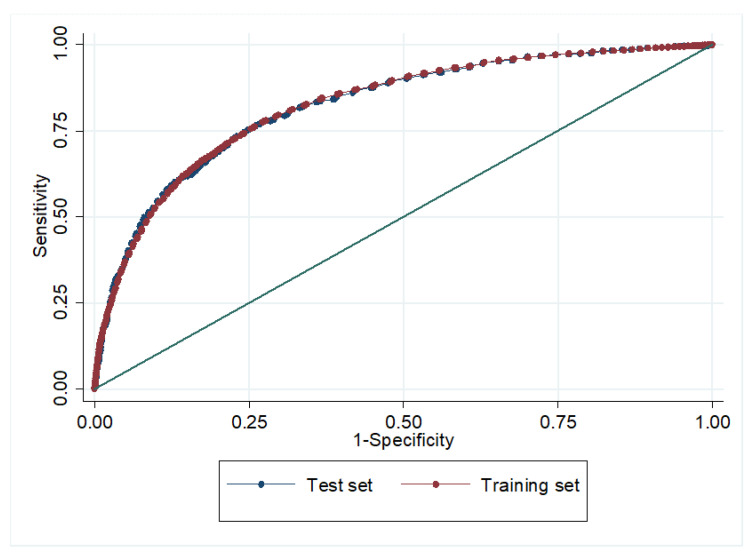
Internal validation. AUC of the model in the training and test sets.

**Figure 3 ijerph-17-04216-f003:**
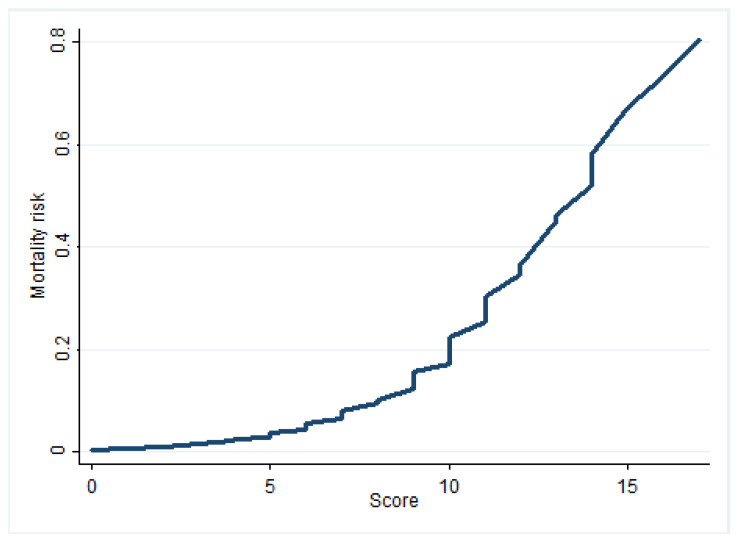
Mortality risk according to the score obtained.

**Table 1 ijerph-17-04216-t001:** Descriptive statistics of the sample for the training set (92,768 episodes).

	Variable	
Quantitative, mean *±* SD	Age	69.56 ± 11.67
	Stay	12.50 ± 6.74
	Preoperative stay	2.51 ± 3.96
	NDD	5.96 ± 3.37
	NPD	3.77 ± 2.94
Qualitative, *n (%)*	Male sex	55,735 (60.10)
	Scheduled admission	71,797 (77.40)
	Re-admission	15,111 (16.30)
	Mortality	3930 (4.20)
	COPD	6301 (6.80)
	Ischaemic heart disease	2608 (2.80)
	Arterial hypertension	38,409 (41.40)
	Obesity	4157 (4.50)
	Stroke	1203 (1.30)
	Renal insufficiency	4019 (4.30)
	Anaemia	10,105 (10.90)
	Atrial fibrillation	7082 (7.60)
	Diabetes	17,210 (18.60)
	Dyslipidaemia	18,826 (20.30)
	Tumor location: rectum	22,363 (24.10)
	Tumor location: sigmoid colon	21,777 (23.50)
	Tumor location: rectosigmoid junction	10,015 (10.80)
	Tumor location: colon/others	38,613 (41.60)

**Table 2 ijerph-17-04216-t002:** Bivariate study for the training set.

Variable		Exitus (3930)	Survival (88,838)	ORu	95% CI	*p*-Value
Age		77.44 ± 10.22	69.22 ± 11.61	1.08	1.08–1.09	<0.001
Stay		14.21 ± 8.77	12.43 ± 6.22	1.04	1.03–1.04	<0.001
Preoperative stay		4.24 ± 6.10	2.44 ± 3.83	1.08	1.07–1.08	<0.001
NDD		9.44 ± 3.53	5.81 ± 3.27	1.34	1.32–1.35	<0.001
NPD		6.41 ± 4.41	3.65 ± 2.80	1.22	1.21–1.23	<0.001
Sex	Female	1428 (3.86)	35,602 (96.14)	1		
	Male	2502 (4.49)	53,233 (95.51)	1.17	1.09–1.25	<0.001
Laparoscopy	No	3848 (4.50)	81,576 (95.50)	1		
	Yes	82 (1.12)	7262 (98.88)	4.17	3.35–5.21	<0.001
Type of admission	Scheduled	1628 (2.27)	70,169 (97.63)	1		
	Urgent	2299 (11.00)	18,609 (89.00)	5.32	4.99–5.68	<0.001
Re-admission	No	3144 (4.05)	74,513 (95.95)	1		
	Yes	786 (5.20)	14,325 (94.80)	1.3	1.20–1.41	<0.001
Tumor location: rectum	No	3399 (4.83)	67,006 (95.17)	1		
	Yes	531 (2.37)	21,832 (97.63)	0.48	0.44–0.53	<0.001
Tumor location: sigmoid colon	No	3020 (4.25)	67,971 (95.75)	1		
	Yes	910 (4.18)	20,867 (95.82)	0.98	0.91–1.06	0.630
Tumor location: rectosigmoid junction	No	3524 (4.26)	79,229 (95.74)	1		
	Yes	406 (4.05)	9609 (95.95)	0.95	0.86–1.06	0.340
Tumor location: descending colon	No	3749 (4.23)	84,956 (95.77)	1		
	Yes	181 (4.45)	3882 (95.55)	1.06	0.91–1.23	0.480
Tumor location: transverse colon	No	3715 (4.18)	85,092 (95.82)	1		
	Yes	215 (5.43)	3746 (94.57)	1.32	1.14–1.51	<0.001
COPD	No	3470 (4.02)	82,949 (95.98)	1		
	Yes	455 (7.22)	5846(92.78)	1.9	1.68–2.06	<0.001
Ischaemic heart disease	No	3711 (4.13)	86,058 (95.87)	1		
	Yes	192 (7.36)	2416 (92.64)	1.84	1.58–2.14	<0.001
Arterial hypertension	No	2392 (4.41)	51,871 (95.56)	1		
	Yes	1534 (3.99)	36,875 (96.01)	0.9	0.84–0.96	<0.010
Valvulopathy	No	3773 (4.16)	86,896 (95.84)	1		
	Yes	138 (7.59)	1679 (92.41)	1.89	1.59–2.26	<0.001
Stroke	No	3781 (4.14)	87,597 (95.86)	1		
	Yes	130 (10.81)	1073 (89.19)	2.81	2.33–3.38	<0.001
Renal insufficiency	No	2807 (3.17)	85,664 (96.83)	1		
	Yes	1014 (25.23)	3005 (74.77)	10.29	9.50–11.16	<0.001
Anaemia	No	3197 (3.88)	79,225 (96.12)	1		
	Yes	714 (7.07)	9391 (92.93)	1.89	1.73–2.05	<0.001
Atrial fibrillation	No	3267 (3.81)	82,380 (96.19)	1		
	Yes	657 (9.28)	6452 (90.72)	2.58	2.36–2.81	<0.001
Diabetes	No	3178 (4.22)	72,188 (95.78)	1		
	Yes	740 (4.30)	16,470 (95.70)	1.02	0.94–1.11	0.630
Obesity	No	3796 (4.28)	84,810 (95.72)	1		
	Yes	133 (3.20)	4024 (96.80)	0.74	0.62–0.89	<0.001
Dyslipidaemia	No	3418 (4.63)	70,353 (95.37)	1		
	Yes	510 (2.71)	18,316 (97.29)	0.57	0.52–0.63	<0.001

Quantitative variables: mean ± SD. Qualitative variables: n (%). ORu: Crude odds ratio.

**Table 3 ijerph-17-04216-t003:** Models of multiple logistic regression for the training set.

	Model 1 (Initial)		Model 2 (Intermediate)		Model 3 (Final or ACMS)	
Variable	ORa	95% CI	ORa	95% CI	ORa	95% CI
Ischaemic heart disease	1.31	1.11–1.56				
Anaemia	1.14	1.04–1.25				
Re-admission	0.72	0.66–0.79	0.72	0.66–0.79		
CRC location Rectum	0.80	0.77–0.89	0.80	0.72–0.88		
Atrial fibrillation	1.40	1.26–1.55	1.39	1.26–1.54		
Dyslipidaemia	0.53	0.48–0.59	0.54	0.48–0.59		
Age (decade)	1.72	1.66–1.79	1.73	1.73–1.79	1.79	1.78–1.79
Type of admission (urgent)	4.65	4.33–5.00	4.63	4.31–4.98	4.68	4.36–5.02
COPD	1.39	1.24–1.56	1.40	1.25–1.56	1.43	1.28–1.60
Stroke	1.90	1.54–2.33	1.89	1.54–2.32	1.87	1.53–2.29
Renal insufficiency	7.17	6.56–7.84	7.18	6.57–7.85	7.26	6.65–7.94
AUC	0.84 95%CI 0.83–0.84		0.83 95%CI 0.83–0.84		0.83 95%CI 0.82–0.83	
Pearson’s χ^2^ test	<0.001		<0.001		<0.001	

ORa: Adjusted odds ratio.

**Table 4 ijerph-17-04216-t004:** Mortality risk score for colorectal cancer.

Attribute	Points
Age 41–60	1
Age 61–80	3
Age > 80	5
Urgent admission	4
COPD	1
Stroke	2
Renal insufficiency	5

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
