# Peer review of "Predictive Model of the Risk of In-Hospital Mortality in Colorectal Cancer Surgery, Based on the Minimum Basic Data Set"

_ijerph, 2020, doi:10.3390/ijerph17124216_

Round 1

Reviewer 1 Report

Thank you for giving me the opportunity to review this manuscript.  The model developed is very important for in-patient mortality rate of colo rectal cancer especially as it relates to the patient admission.

The introduction is well done and informative.  There are however various areas where acronyms are used without some basic description for the reader (see line items 40-60).  

Materials and Methods:  Please review starting the sentence for grammatical correctness.  On line item 99 you mentioned using a moderate outlier detection procedure.  I recommend to please briefly describe it for the reader. 

Results: I was wondering if the study results were compared in terms of other in-hospital risk adjusted mortality rate; or perhaps with other similar hospitals. Also, was there a time frame for the mortality rate? I was not able to see it.  In terms of the calibration, for internal and external survival probabilities was there a time frame?

Discussion and limitations were well done. 

Author Response

Dear Editor. Dear reviewer,

We appreciate the effort for all the suggestions to improve and clarify the manuscript, as well as the detailed analysis of the reviewers.

We provide two point by point responses to each of the issues raised by the reviewer. These responses have resulted in changes within the manuscript (for clarity of the document) that have been made using Word change control. They have also been referred to the line number in the text.

On the other hand, the quality of the text has been re-evaluated and improved in the English version by a native translator. For this reason, you will find, in addition to the paragraphs we have cited, some other changes that respond to improvements in English (also made with change control).

In the hope and trust that the modifications made and changes introduced will satisfy the questions raised, we are enclosing all the new documentation.

We remain at your disposal for any other changes that may be necessary.

Best regards

Juan Manuel García Torrecillas

Reviewer 2 Report

Journal: IJERPH

Manuscript ID: ijerph-782517-peer-review-v1

Date: 2020/05/23

Title: Predictive model of the risk of in-hospital mortality in colorectal cancer surgery, based on the Minimum Basic Data Set

Authors: García-Torrecillas Juan Manuel et al.

GENERAL COMMENTS (DO NOT NEED TO BE ANSWERED).

The authors use a retrospective cohort (using an administrative hospital discharge database) of 115,841 patients admitted for colorectal cancer surgery between 2008 and 2014 in Spain, to develop a predictive score of in-hospital mortality, and also they carried out an internal validation of the model (using a development and a validation databases).

MAJOR CONCERNS AND COMPULSORY REVISIONS

  1. While the authors say (objectives, page 2, lines 70-72) that they will generate “a score based on variables present at the time of patient admission”, the MBDS collects events that occur throughout the whole hospitalization (not only those present at the time of admission) including the surgical complications themselves. Therefore, the developed score is not so much a "predictive model" that can be used at patient admission, but rather a score that predicts mortality with the information available up to the moment of death (which may have some utilities, but not the potential uses indicated in the paper as “The proposed SCMS model has the advantage of predicting the risk of mortality for patients diagnosed with CRC before any surgery takes place, and therefore suitable measures can be adopted in good time”.
  2. Reinforcing the previous comment, digestive bleeding, ileum, suture dehiscence, renal failure, aspiration pneumonia, etc., are used as "predictor" variables, without the MBDS allowing to differentiate whether they were present on admission or were -as is likely in many cases- complications of surgery that increased the risk of death, and perhaps they should be part of a combined dependent variable (severe complication or death) rather than be used as predictive variables. It also explain the high predictive value of some of these complications in the model and the high discrimination capacity.

MINOR ESSENTIAL REVISIONS

  1.  

DISCRETIONARY REVISIONS

  1. None

Author Response

(The authors gave the same response as above.)

Round 2

Reviewer 1 Report

Thank you for the opportunity to review once more.  The manuscript is much improved.  There are no additional recommendations.

Thank you.

Author Response

We appreciate all suggestions sent by the reviewer. Your comments have helped, in an important way, to improve the quality and contents of the manuscript. Thank you.